# Desmin disorganisation: A key feature in feline hypertrophic cardiomyopathy

Wan-Ching Cheng[1], Charlotte Lawson[2], Lois Wilkie[1], Melanie Dobromylsky[3], Virginia Luis Fuentes[1], Mark R. Holt[4], Elisabeth Ehler[4‡], David J. Connolly[1‡*]

1 Department of Clinical Science and Services, Royal Veterinary College, Hatfield, Hertfordshire, United Kingdom, 2 Department of Comparative Biomedical Sciences, Royal Veterinary College, Royal College Street London, United Kingdom, 3 Finn Pathologists, Harleston, Norfolk, United Kingdom, 4 British Heart Foundation Centre of Research Excellence, School of Cardiovascular and Metabolic Medicine and Sciences, King's College London, London, United Kingdom

‡ EE and DJC are joint senior authors on this work.
* dconnolly@rvc.ac.uk

## Abstract

Hypertrophic cardiomyopathy is usually characterised histologically by increased ventricular wall thickness and myocyte disarray. In human and rodent HCM, subcellular alterations were detected that involve the intermediate filament cytoskeleton (mainly desmin) and proteins that are important for mechanical and electrochemical connection of the cardiomyocytes (beta-catenin and connexin-43, respectively). We demonstrate here that similar changes can be visualised in HCM samples from cats, with prominent desmin and αB-crystallin aggregates that are accompanied by increased expression at the protein level. In addition, there is a disorganisation of beta-catenin and connexin-43, which display additional aberrant signals at the lateral surface of cardiomyocytes. This suggests that the subcellular response in cardiomyocytes to HCM is shared by humans and cats.

## Introduction

Hypertrophic cardiomyopathy (HCM) is a serious disease in humans and cats and exhibits considerable similarities at the molecular, cellular and whole organ levels [1–5]. It is characterized by a hypertrophied and non-dilated left ventricle in the absence of abnormal loading conditions capable of producing left ventricular hypertrophy [6]. HCM is the most common heart disease in cats affecting up to 15% of the general feline population [7]. Echocardiography is the clinical gold standard for HCM diagnosis in cats, and the most used imaging modality in humans [6,8]. Clinical signs in cats include respiratory distress due to congestive heart failure, pelvic limb ischaemic paralysis/paresis due to aortic thromboembolism or sudden arrhythmogenic death. Prognosis for cats with HCM is very variable, with median survival times ranging from 596 to 1276 days [9]. The role of genetic testing is very limited in cats as to date only two disease associated variants in MYBPC3 have been identified, one in the Maine coon and one in

**Data availability statement:** All relevant data are within the manuscript and its Supporting Information files.

**Funding:** Dr Wan-Ching Cheng receive funding (Stipend and tuition fees for her PhD) from the Taiwan Ministry of Education.

**Competing interests:** The authors have declared that no competing interests exist.

the Ragdoll breed; additionally, a potentially pathogenic variant was found in MYH7 in a domestic shorthair [10]. Histological features include cardiomyocyte hypertrophy and disarray, interstitial fibrosis and intramural vascular pathology [4,11,12].

Additionally in cardiomyocytes from human HCM patients, cytoplasmic aggregates of the intermediate filament (IF) desmin and its chaperone heat shock protein αB-crystallin together with disorganisation of associated junctional proteins including β-catenin, N-cadherin and connexin43 were identified [13–17]. Furthermore, a comparable pattern of cytoplasmic pathology has been documented in a variety of other cardiac conditions [17–19] including cardiomyopathies resulting from genetic mutations in desmin and junctional proteins [20–29]. The development of such aggregates is suggestive of IF disorganisation and inadequate protein quality control (PQC) which underlies the cells' inability to completely remove the toxic accumulation of malformed proteins and suggests that IF disruption represents a shared cellular response to a variety of genetic and acquired cardiac insults. There is a sparsity of information describing IF configuration in feline cardiomyocytes or the impact of HCM on their spatial arrangement. However, in one study cytoplasmic aggregates of unknown composition were detected in cardiomyocytes from cats with HCM and in a second study a variable pattern of staining for desmin was identified in HCM cats compared to controls but cytoplasmic aggregates were not seen [2,30].

Desmin is the most abundant IF protein in cardiomyocytes, it extends from the nucleus to the cell membrane where it associates with costameres, the lateral anchorage for cardiomyocytes to the extracellular matrix and to desmosomes and area composita at the intercalated disc (ID), highlighting its role in maintaining cell adhesion, intercellular communication and electrical coupling [29,31–33]. Refer to Fig 1 for the arrangement of desmin, αB-crystallin, and the junctional proteins in healthy cardiomyocytes.

Desmin is also critical for mechanotransduction between cardiomyocytes and the extracellular matrix as well as between adjacent cardiomyocytes [34–37]. Additionally, its interaction with mitochondria assists with their alignment, morphology, and function [32,35,36,38–41]. Therefore, disorganisation of desmin and its interaction partners could contribute to important pathological features of HCM including mitochondrial dysfunction, resulting in energy imbalance, cardiomyocyte death, fibrosis and arrhythmogenesis [23,32].

In light of the disruption to the IF network and junctional proteins described in humans with HCM [13–17] we sought to identify and further characterise the localisation of these proteins in a cohort of cats with HCM. Specifically, we assessed the quantity of desmin and described its cellular organisation in cardiomyocytes from the left ventricular free wall (portion of the left ventricle wall that is not connected to the interventricular septum or the cardiac apex) of cats with and without HCM. Additionally, we explored the effect of desmin disorganisation on the cellular location of other closely associated proteins including (1) αB-crystallin, a chaperone for desmin, (2) β-catenin, a component of the adherens junction, and (3) connexin43, a gap junction protein and found this to be disturbed in feline HCM.

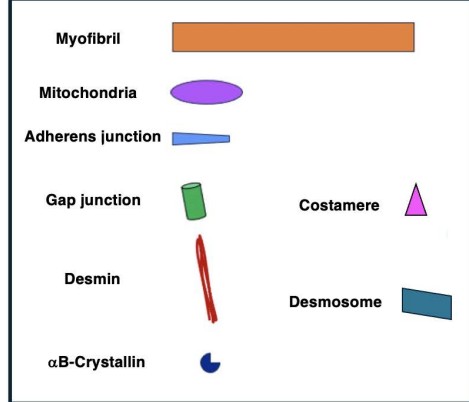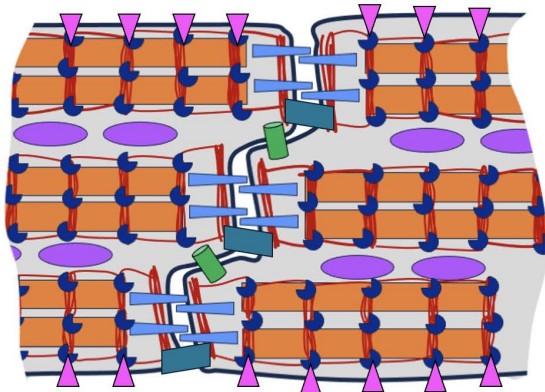

**Fig 1. Schematic illustration summarising the arrangement of desmin, αB-crystallin, and the junctional proteins in normal cardiomyocytes.** In the normal heart, desmin co-localises with αB-crystallin and is aligned into stripes at the Z-discs and intercalated discs. The junctional proteins including β-catenin and N-cadherin which form adherens junctions, and connexin43 which form gap junctions, are situated at the intercalated disc and co-localise with desmin.

## Materials and methods

### Ethics statement

This project has been ethically reviewed by the University Clinical Research Ethical Review Board (CRERB). CRERB reviews all clinical research proposals involving both animals and human subjects and has granted ethical approval (URN 2016 1485) for the project. The animals used in this study were clinical cases not experimental animals. Euthanasia was performed on humane ground to alleviate suffering following discussion between owners and clinicians using an overdose of intravenous barbiturate. Written consent was obtained for euthanasia following normal hospital policy and for cardiac material to be used after death for the study. No personal data related to the owners of the cats was recorded or stored for this study. Only the cats case number, signalment, history and clinical findings were recorded and stored.

### Study population

Cats were recruited from the Royal Veterinary Collage Small Animal referral or first opinion hospitals. Control group cats were those that died of non-cardiac disease with no cardiac related abnormalities detected on clinical history or by clinical exam and gross/histopathological examination. Complete echocardiographic examination or Cardiac TFAST (a rapid thoracic ultrasound examination used in an emergency situation with the animal generally in sternal recumbency optimising the image to evaluate cardiac, pulmonary, mediastinal and pleural structures) was performed in 11/12 control group cats. HCM group cats were those that showed clinical signs compatible with HCM (congestive heart failure, aortic thromboembolism, arrhythmia, gallop sound, murmur) and confirmed by complete echocardiographic examination or Cardiac TFAST and gross/histopathological examination. The HCM cats had no other disease that would result in left ventricular hypertrophy including hypertension, hyperthyroidism, congenital cardiac disease, or infiltrative disease. Detailed information on signalment, echocardiographic parameters and histopathological diagnosis are given in S1 Table (supporting information). Refer to S2 Table for clinical presentation, reason for euthanasia and a summary of cardiac imaging. Left ventricular samples from a total of 12 control and 23 HCM cats were used for evaluation of protein expression. Full details of which cats were used for Western blotting, which for immunohistochemistry and which for both modalities are given in S1 and S2 Tables. In brief 3/12 control and 4/23 HCM cats were used for only Western blotting, 2/12 control and 13/23 HCM cats were used for only immunohistochemistry and 7/12 control and 6/23 HCM cats were used for both Western blotting and immunohistochemistry.

## Cardiac imaging

Cardiac imaging was performed as previously described [42]. In brief, measured echocardiographic parameters included left atrium to aortic ratio (LA/Ao), maximal left ventricular free wall thickness in diastole (LVFWd), maximal interventricular septum thickness in diastole (IVSd), presence of systolic anterior motion of the mitral valve (SAM), presence of spontaneous echo contrast (SEC) and presence of a formed thrombus in the LA or left auricular append-age (S3 Table). HCM was defined as a diastolic left ventricular wall thickness (LVFWd or IVSd) measuring ≥6 mm on 2-dimensional (2D) imaging [43] with or without papillary muscle hypertrophy, SAM or dynamic left ventricular outflow tract obstruction (LVOTO) [44]. All echocardiographic examinations were performed by a veterinary cardiology special-ist or resident under direct supervision. Point of care ultrasound (POCUS) examinations were performed in the emergency setting by a veterinary emergency and critical care specialist or resident in training under direct supervision and facilitated measurement of LA/Ao as described above and a subjective assessment of left ventricular wall thickness in diastole.

## Heart collection and histopathological examination

The heart was harvested and the residual blood in the heart was flushed with tap water within 30 minutes of euthanasia. A 10 x 10 mm full thickness left ventricular (LV) free wall segment was excised and preserved in RNAlater (Qiagen). The remainder of the heart was immersed in a 10% formaldehyde solution for 24 hours before undergoing dissection and processing. Four-micron thickness sections were prepared from the paraffin-embedded cardiac tissues and stained with either haematoxylin and eosin or Masson's Trichrome stain. Histopathological diagnosis was made by a specialist veterinary pathologist (MD, LW) [45]. Macroscopic indicators for HCM comprise left ventricular wall thickening with or without enlargement of the left atrium or both atria [46]. Microscopic indicators for HCM encompass more than 5% of the left ventricle displaying myofiber disarray, with or without myocyte enlargement, interstitial fibrosis, or intramural arteriosclerosis [45,47].

## Immunostaining

Cardiac tissues in paraffin blocks were cut into a 4-micrometre section and de-paraffinised and rehydrated using xylene, ethanol of 100%, 90%, 70% and 30% concentration, and double distilled $H_2O$. Antigen retrieval was accomplished by incubating the slides in 10mM pH6 citric acid buffer at 92.5°C for 15 minutes. Sections for chromogenic immunostaining were first blocked with BLOXALL (Vector Laboratories) to quench the activity of endogenous peroxidase for 15 minutes at room temperature. Sections were blocked with either 2.5% normal horse serum (Vector Laboratories) for chromogenic immunostaining or 3% normal goat serum (Sigma) for 30 minutes at room temperature depending for fluorescent immunostaining. Sections were then incubated with the primary antibody (β-catenin, Sigma-Aldrich, C2206; β-catenin, Santa Cruz Biotechnology, E-5, sc-7963; N-cadherin, Santa Cruz Biotechnology, D-4, sc-8424; Connexin43, ThermoFisher, 71–0700; Desmin, Agilent, M0760, M076029-2) at 4°C overnight followed by incubation with horseradish peroxidase (Vector Laboratories) for chromogenic immunostaining; or DAPI (Sigma), fluorophore conjugated anti-rabbit or anti-mouse secondary antibodies (multilabelling quality; Jackson ImmunoResearch) for fluorescent immunostaining, at room temperature for an hour. Double immunofluorescence stainings were always performed with two antibodies raised in different species, except for desmin (monoclonal mouse antibody) and αB-crystallin (αB-crystallin conjugated with Alexa Fluor® 488, Santa Cruz Biotechnology, F-10, sc-137129; monoclonal mouse antibody) where indirect immunofluorescence staining was performed first for desmin followed by direct immunofluorescence staining for αB-crystallin. For chromogenic immunostaining, 3,3'Diaminobenzidine (DAB) substrate (Vector Laboratories) was used for colour development and the slides were counterstained with haemotoxylin. All slides were then mounted, coverslipped, and sealed before examination. The number of cats used for immunostaining for each protein is listed in Table 1.

**Table 1. Number of cats per group used for immunostaining.**

| Protein | Control (N) | HCM (N) |
|---|---|---|
| Desmin | 9 | 18 |
| αB-crystallin | 5 | 5 |
| β-catenin | 4 | 6 |
| N-cadherin | 3 | 3 |
| Connexin43 | 3 | 8 |

## Image analysis

Fluorescent labelling of the proteins was detected using a Leica DMRA2 upright microscope connected to a monochrome camera (AxioCam). A Leica DM4000B with DFC550 colour microscopy camera (Leica) was used to take microphotographs for chromogenic IHC. The microscopes and cameras were controlled using the Leica Application Suite Version 4.12. Slides were examined under 200 or 400x magnification and the detection of overexposure facility was turned on during observation.

## Western blotting

Protein lysates from the mid LV free wall were used for Western blotting. Homogenised cat LV free wall (12 µg) was electrophoresed on precast gradient gel (Bolt 4−12% Bis-Tris Plus Gels, ThermoFisher) and transferred to polyvinylidene difluoride membrane (Pierce PVDF Transfer Membrane 0.45 µm, ThermoFisher) which was later blocked with 5% milk (0.1% fat dry milk, Marvel) in Tris-buffered saline 0.1% Tween 20 (TBST) followed by an overnight incubation in agitation with the primary antibodies (Desmin, Agilent, M0760, M076029-2; Desmin, R&D, AF3844; αB-crystallin, Santa Cruz Biotechnology, F-10, sc-137129; β-catenin, Sigma-Aldrich, C2206; GAPDH, Novus Biologicals, 1D4, NB300−221) at 4°C, a 1-hour incubation with horseradish peroxidase-conjugated secondary antibodies (Goat anti-rabbit IgG, Pierce, 1:1500; Goat anti-mouse IgG, Pierce, 1:1500), and enhanced chemiluminescence substrate (Western Lightning Plus-ECL, Perkins Elmer) sequentially. All the antibodies were suspended in 0.5% milk in TBST. The membrane was washed three times in TBST between each incubation. The developed signals were detected using ChemiDoc MP Imaging System (Bio-Rad) and densitometry (ImageJ) with GAPDH as loading control was performed to semi-quantified protein level. Five control and 5 HCM cats were used for immunoblotting of desmin and αB-crystallin; and 10 control and 10 HCM cats for used for junctional protein β-catenin immunoblotting. αB-crystallin was analysed in the same set of cats as used for desmin to assess paired expression. β-catenin was analysed using membrane resources from a parallel study on myocardial fibrosis [42], which included five additional cats in each group.

## Image processing and cepstral analysis with circular variance calculation

Each image was imported from a predefined list of files into Wolfram Mathematica 13 (Champaign, Il). The images were first adjusted for contrast using the ImageAdjust function, and subsequently converted to grayscale using ColorConvert. To apply a pixel-wise transformation based on spatial location, a shift operation was conducted. This transformation was defined by a non-linear shift in pixel intensity, proportional to the Euclidean distance from the center of the image, scaled by the image dimensions. The resulting transformed image was clipped to ensure pixel values remained within the normalized range [0, 1]. After transformation, a Gaussian filter was applied with a kernel size of 2, and the filtered image was subtracted from the original. This created a sharper image for further analysis. The cleaned image was then subjected to cepstrogram analysis. The cepstrogram was computed by cropping the image to central regions, followed by the calculation of the absolute magnitude of the Fourier periodogram followed by obtaining the absolute magnitude of the inverse Fourier transformation to create the cepstrogram. A polar transformation of the cepstrogram was performed to analyze the

circular symmetry of cepstral components. The transformation was calculated using angular coordinates, converting the cartesian image coordinates to polar coordinates. A line plot of the cepstrogram data was generated to visualize the mean cepstral coefficients. Circular variance, a measure of the dispersion of angular orientations in the image, was computed from the orientation data extracted from the polar-transformed cepstrogram. The position of the maximum orientation value was identified and the mean values of pixel intensities at different angular positions were used to calculate the circular variance, using the formula:

$$1 - \frac{\sqrt{\text{Total}[\sin(\text{angle})]^2 + \text{Total}[\cos(\text{angle})]^2}}{l}$$

## Statistical analysis

Continuous data are reported as mean (SD) or median [interquartile range or range] depending on the result of a Shapiro-Wilk normality test. Student t test, or Welch t test was used if the equal variance assumption was violated to detect difference in age. Mann-Whitney test was used to analyse protein expression between groups. Categorical data are reported by percentage and were analysed using Fisher's exact test. Correlation was assessed using Spearman's rho test. Two-tailed analysis was used and the results were considered significant when $p < 0.05$. All analyses were performed using commercial software (GraphPad Prism 8, GraphPad Software).

## Results

### Animals

The median age (range) was 6.3 (1.5–19.0) years in the control group and 7.0 (1.7–15.0) years in the HCM group ($p > 0.7$). Three control cats were known to be less than 3 years old although their exact age was unknown. A presumptive age of 1.5 years was used for these three cats to perform statistic tests. As the age range was large, we also assessed the percentage of cats >7 or ≤ 7 years in both groups. In the control group 41.7% (5/12 cats) were > 7 years, in the HCM group 47.8% (11/23 cats) were > 7 years. Male cats accounted for 41.7% (5 out of 12) and 73.9% (17 out of 23) of the cats in the control and HCM group, respectively ($p = 0.07$). A full echocardiographic examination was performed in 3/12 control and 14/23 HCM cats. Cardiac focused TFAST imaging was performed in 8/12 control and 9/23 HCM cats.

### Increased expression and disorganisation of desmin and αB-crystallin in HCM cats

Because cytoplasmic aggregates of desmin and αB-crystallin were documented in human cardiac patients and rodent models of heart disease we sought to examine the cellular localisation of desmin and αB-crystallin and assess their spatial relationship using immunohistofluorescence in feline hearts. In the control cats, desmin localised to the Z-discs and intercalated discs as fine and thicker stripes respectively in the cardiomyocytes. In contrast, in the tissues from HCM affected hearts, the desmin signal was frequently lost from the intercalated discs with or without preserved desmin organisation at the Z-disk. In addition, a severely disorganised pattern of desmin, characterised by a complete loss of the typical stripe pattern was seen in some cardiomyocytes in all HCM cats that were analysed (Fig 2).

In light of our immunohistofluorescence data and the proposed role of impaired protein quality control (PQC) in HCM, we quantified the amount of desmin and its chaperone protein αB-crystallin in the myocardium of cats with and without HCM using Western blot. Desmin and αB-crystallin were identified as bands at 50kDa and 22kDa respectively. The HCM cats expressed increased amounts of desmin (median 1.41 [range 0.59–2.19] (N = 5) versus 0.54 [0.27–0.69] (N = 5); $p = 0.016$) and αB-crystallin (median 1.46 [range 0.63–2.39] (N = 5) versus 0.56 [0.18–0.66] (N = 5); $p = 0.032$) compared to the controls (Fig 3).

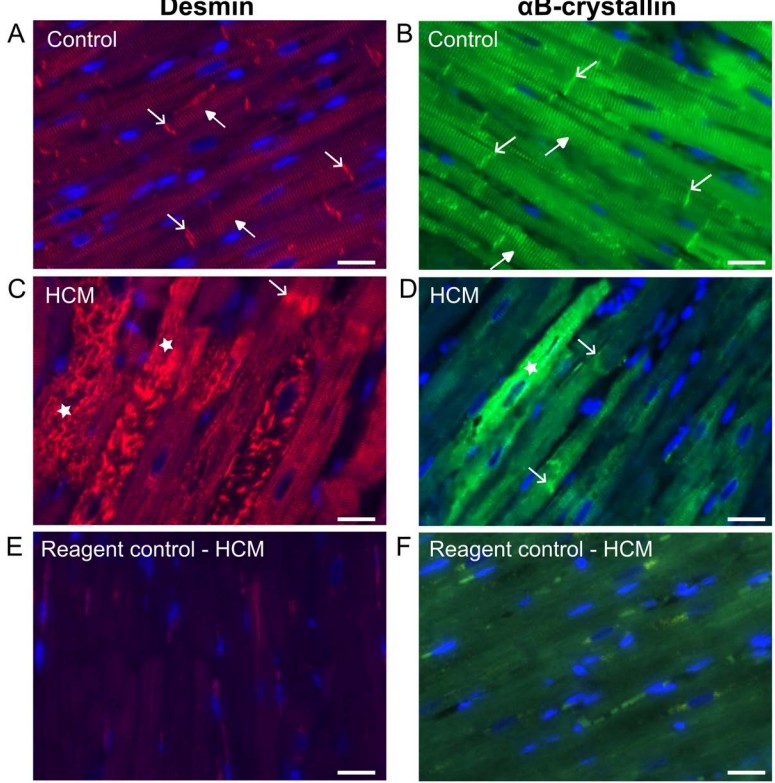

**Fig 2. Aberrant localisation of desmin and αB-crystallin in feline HCM samples. A, B Confocal micrographs of feline hearts immunostained for desmin and αB-crystallin.** In the control cats, desmin and αB-crystallin located to the Z-discs (closed arrow heads) and intercalated discs (open arrow heads) in the cardiomyocytes. C, D In the HCM cats, disorganised and aggregated desmin and αB-crystallin (star) were seen in some of the cardiomyocytes. Immunostaining of desmin and αB-crystallin varied in intensity between cardiomyocytes with some cell staining with hight intensity (bright red/green) or lower intensity (duller red/green). The degree of desmin and αB-crystallin loss at the Z-discs and intercalated discs also varies between different cardiomyocytes. Open arrowheads indicate the preserved though less distinct desmin and αB-crystallin at the intercalated disc. E, F Reagent control was performed by omitting the primary antibody. 400x magnification; Desmin – red; αB-crystallin – green; Nuclei – blue; Scale bar = 25 μm.

## Correlation of desmin and αB-crystallin

Since αB-crystallin is an important chaperone protein for desmin, the association between these two proteins was assessed. Quantification by Western blot indicated that desmin and αB-crystallin expression levels were highly correlated (Fig 3C).

We then explored the localisation of αB-crystallin and desmin using double immunostaining to ascertain whether the two proteins were spatially associated. αB-crystallin showed overlapping distribution with desmin in both control and HCM cats. Increased αB-crystallin fluorescence was observed in regions of elevated desmin signal, particularly within cytoplasmic aggregates in cardiomyocytes from HCM cats (Fig 4). In addition, there was evidence of autonomous increase in labelling of both proteins given the variation in overlay colour in the labelled sections. This would suggest that expression of both proteins is regulated independently.

## Two dimensional cepstral analysis of LV tissue sections from control and HCM cats

Given the disorganisation of desmin in the HCM cats identified by immunoblotting we performed two dimensional cepstral analysis on LV sections from (7 control cats) and (16 HCM cats) as a way of further determining the desmin organisation in cardiomyocytes from diseased cats (Fig 5). Images from regions with minimal aggregates were used

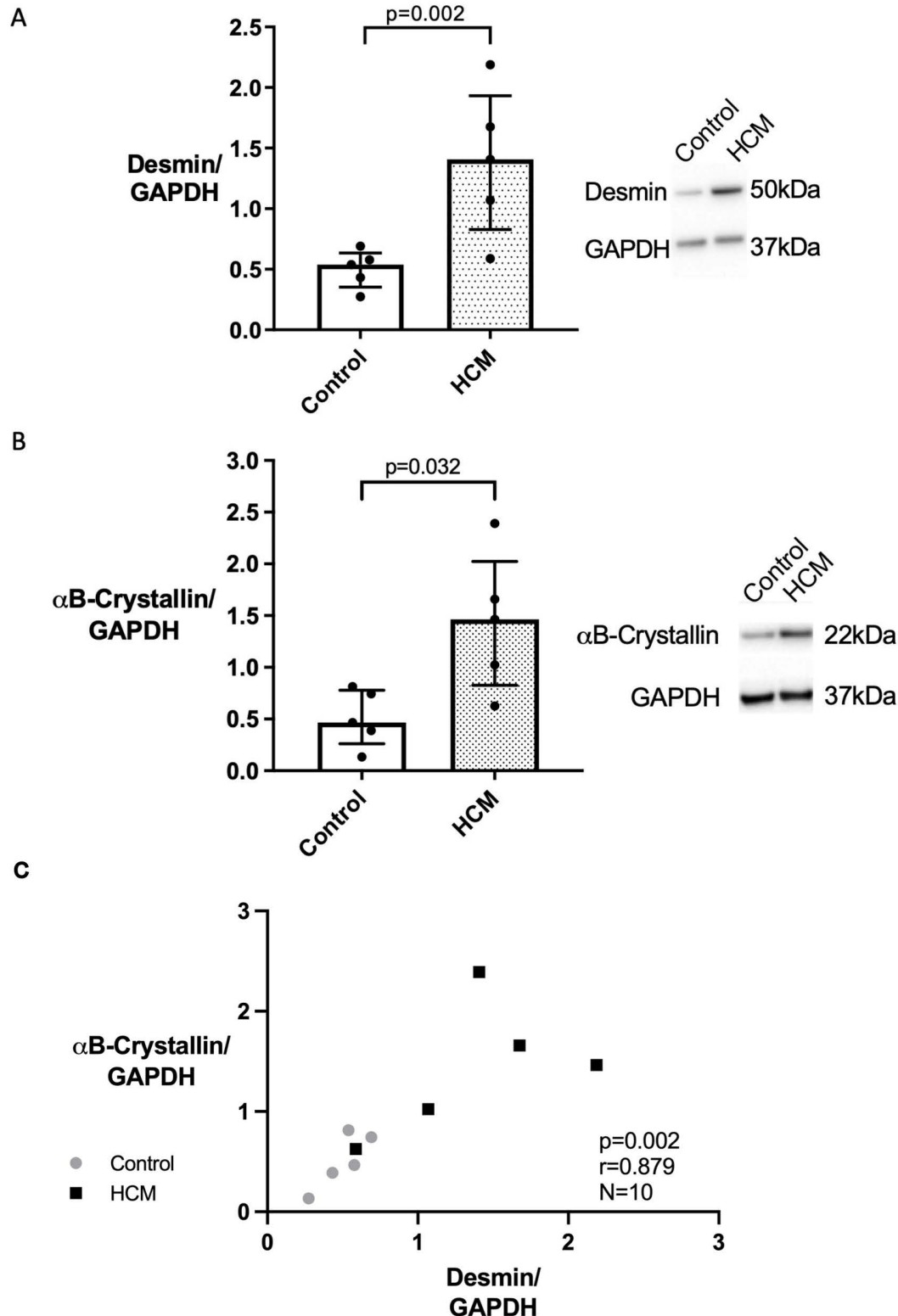

**Fig 3. Desmin and αB-crystallin expression is increased in feline HCM with a strong positive correlation between the expression of both proteins.** Immunoblotting on whole heart extracts from control (white bar; n=5) and HCM (dotted bar; n=5) cats was carried out using a mouse mono-clonal antibody against desmin (panel A) and a mouse monoclonal antibody against αB-crystallin (panel B). Bands at the expected molecular weight (50

and 22 kDa, respectively) were detected using enhanced chemiluminescence. GAPDH was used as a loading control. The right shows a representative blot, while quantifications are shown on the left. A Desmin was significantly increased in the HCM compared to the control samples. B αB-crystallin was significantly increased in the HCM compared to the control samples. Data were analysed with Mann-Whitney test and graphed in bar chart format showing individual data points, median, 25th and 75th quartiles. C A strong correlation was seen between Desmin and αB-crystallin expression quantified by immunoblotting. Data was analysed using Spearman's correlation test.

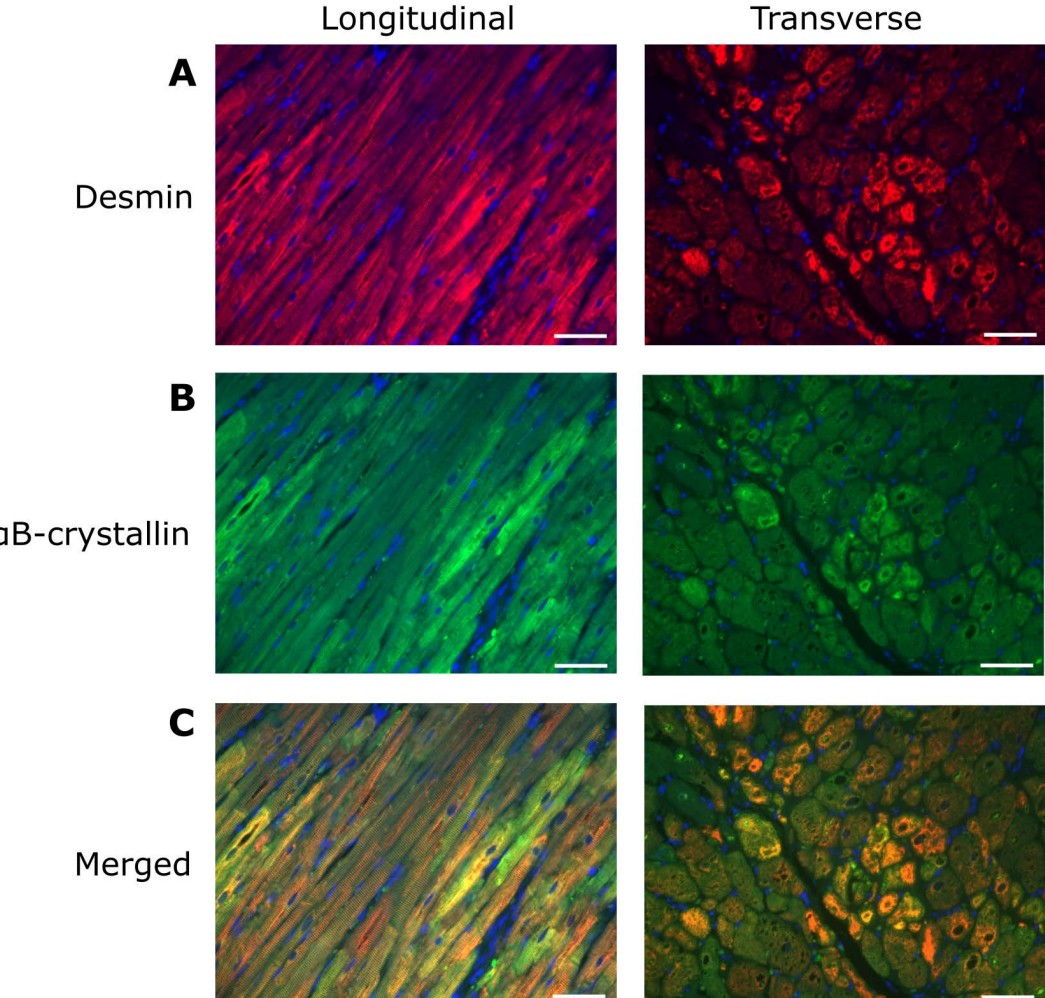

**Fig 4. Cytoplasmic aggregation of desmin and αB-crystallin within individual cardiomyocytes in HCM hearts.** Images from a HCM cat with fluorescent signals from individual channel for desmin and αB-crystallin are displayed in A and B. Merged images in C showed co-localisation of desmin and αB-crystallin in yellow. Increased expression of desmin coincided with increased αB-crystallin expression particularly within cytoplasmic aggregates. There is also evidence of autonomous increase in labelling of both proteins since in addition to yellow labelling consistent with co-localisation of the two proteins, cells with primarily green and orange/red labelling was also observed. 400x magnification; Desmin – red; αB-crystallin – green; Nuclei – blue; Scale bar = 50 μm.

for analysis in order to assess the arrangement of desmin at Z-discs. Our analysis confirmed a greater degree of desmin disorganisation in LV samples from HCM cats compared to control cats in agreement with our immunostaining results.

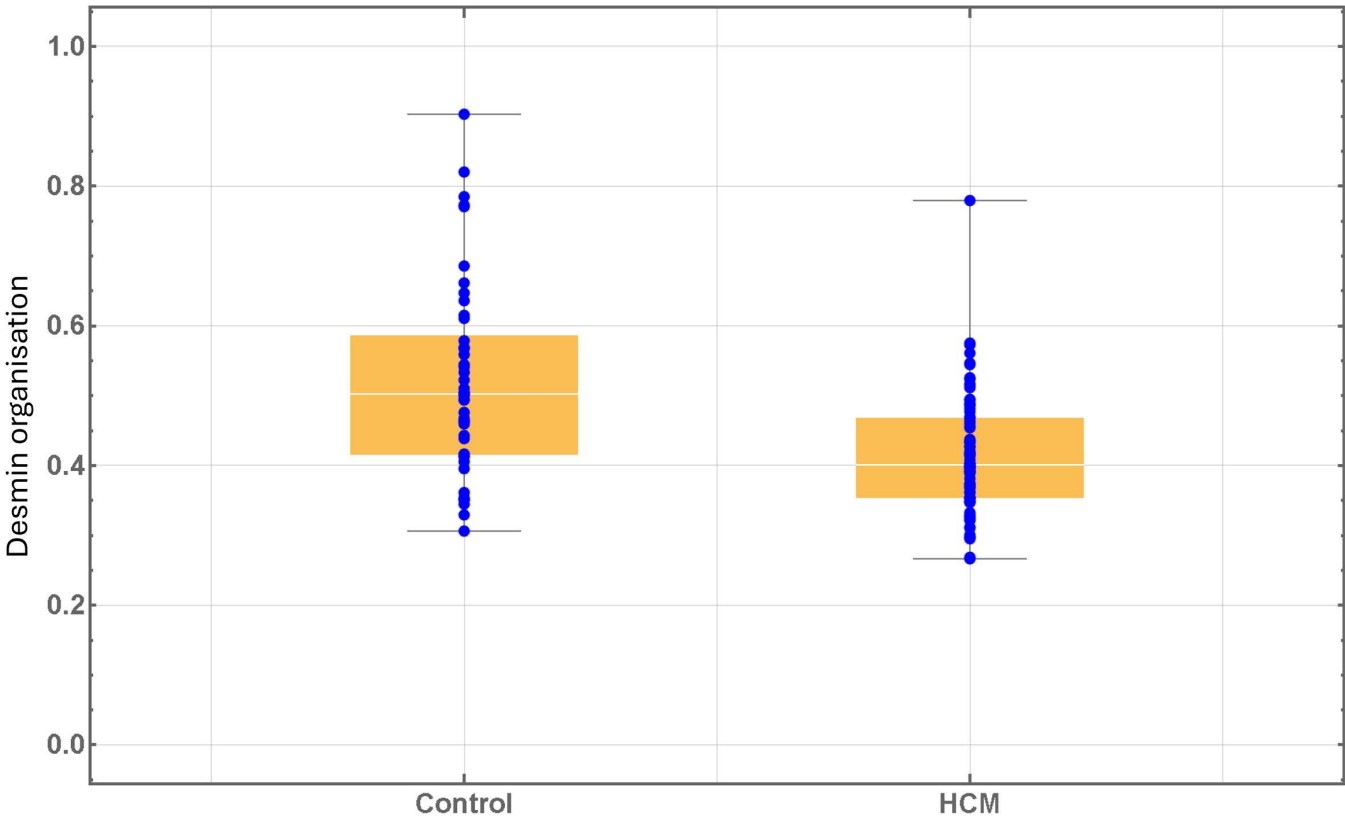

**Fig 5. FFT Two-dimensional cepstral analysis of left ventricular tissue sections from control and HCM cats.** More desmin disorganisation was noted in the HCM sections compared to control sections (p<0.001). The scale is 0-to-1 where 0 is completely disordered and 1 is completely ordered.

## Disorganisation of adherens junction proteins

Alteration in the cellular localisation of key proteins associated with the ID including those forming the adherens junctions were identified in humans with hypertrophic cardiomyopathy [16,17,29]. We therefore examined the cellular localisation of the adherens junction proteins β-catenin and N-cadherin in control and HCM cats using immunohistofluorescence (Fig 6). β-catenin localised to the intercalated discs as expected in the control group (N = 4). As described for human cardiomyopathy, aberrant location of β-catenin, usually to the lateral side of the cardiomyocytes, was seen in 5 of the 6 HCM cats (N = 6) but not observed in any of the controls. A clear association between aberrant localisation of β-catenin and feline HCM was identified (p = 0.048).

Double immunostaining for desmin and β-catenin revealed that in some cats with HCM, desmin staining could be lost at the ID while β-catenin localisation at the ID remained unaffected. However, in HCM cats with more severe disruption of desmin architecture, staining for both desmin and β-catenin could be lost at the ID (Fig 7).

Since we exclusively used paraffin embedded sections for this work, we confirmed our findings by chromogenic immunohistochemistry of β-catenin as this is the standard protocol used in human histological studies. We again identified abnormal expression of β-catenin in HCM cats by immunostaining, either as a diffuse pattern within the cytoplasm or displaced laterally to the cell periphery (Fig 8). Additionally, as N-cadherin also forms part of the adherens junction with β-catenin, we assessed the localisation of N-cadherin in 3 control and 3 HCM cats with immunohistochemistry and showed that the pattern of expression of N-cadherin in both the control and HCM cats resembled that of β-catenin (Fig 8).

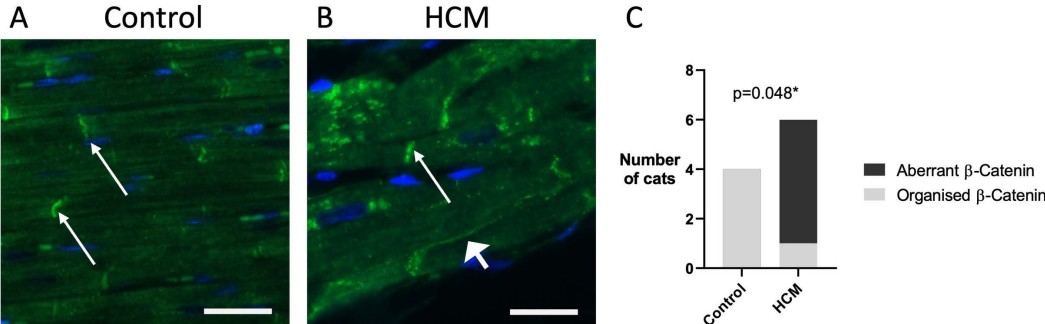

**Fig 6. Abnormal localisation of β-catenin protein in feline HCM samples.** A β-catenin located at the intercalated discs (thin arrow) in the control cats. B In HCM cats, β-catenin was observed at the intercalated discs (thin arrow), in addition, β-catenin showed an aberrant localisation usually seen at the lateral side (thick arrow) of cardiomyocytes. C A bar chart shows the association of feline HCM and aberration of β-catenin localisation (p=0.048). Data was assessed using Fisher's exact test. 400x magnification. β-catenin – green; Nuclei – blue; Scale bar = 25μm.

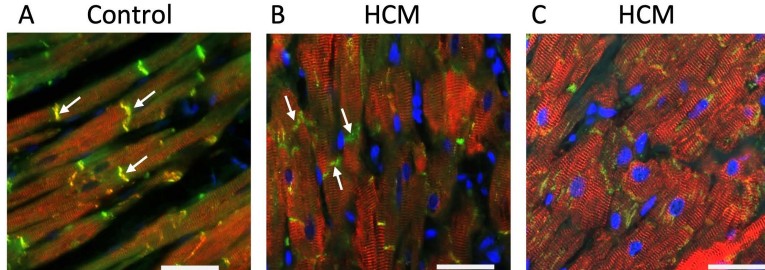

**Fig 7. Double immunostaining for desmin and β-catenin protein revealed normal localisation of β-catenin at the intercalated discs in the presence of desmin disorganisation.** A Desmin (red) and β-catenin (green) co-localised (yellow) at the intercalated discs (arrows) in control samples. B Desmin disorganisation with reduced labelling at the intercalated discs while β-catenin labelling (green) was relatively unaffected (arrows). C More severe desmin disorganisation away from intercalated discs caused further reduction in co-localisation, (less yellow labelling) while β-catenin labelling (green) remained present at the intercalated discs (arrows). 400x magnification. Desmin – red; β-catenin – green; Nuclei – blue; Scale bar = 25μm.

Given the variation in expression of β-catenin by immunostaining in the HCM cats, we then quantified its expression using immunoblotting. On Western blot, β-catenin was identified as a band at 92kDa and a significantly greater quantity of β-catenin was identified in the HCM cats compared to controls (median 1.23 [IQR 0.81–1.64] (N = 10) versus 0.79 [0.64–0.97] (N = 10); p = 0.044) (Fig 9).

### Disorganisation of connexin43

Given the known association between desmin and connexin43 [48,49] and the altered localisation of connexin43 in human cardiomyopathy patients [15,29], we then compared the cellular distribution of connexin43, the building block of the gap junction in control cats and those with HCM. As expected, connexin43 was consistently expressed at the intercalated discs in the control cats. In contrast, in the HCM cats there was reduced expression of connexin43 at the ID and significant peripheral staining along the lateral sides of the cardiomyocytes. The localisation of connexin43 was significantly different between control and HCM cats (p = 0.024) and aberrant localisation was seen in 7 (out of 8) HCM cats but not in the control cats (Fig 10).

We then performed double immunostaining to examine the spatial relationship between desmin and connexin43. The lateralisation of connexin43 and partial loss of expression from the ID was seen even when desmin was still expressed at

# β-catenin

# N-cadherin

**A** Control

**B** Control

**C** HCM

**D** HCM

**Fig 8. β-catenin and N-cadherin both show aberrant localisation in feline HCM.** β-catenin and N-cadherin were only observed at the intercalated discs (open arrowhead) in the control cats (A, B) while in the HCM cats (C, D) β-catenin and N-cadherin was frequently absent at the intercalated discs or assumed a diffuse pattern within the cytoplasm (open arrow heads). Lateralisation of the β-catenin is shown by the closed arrowhead, however, lateralization of N-cadherin was not observed in the 3 HCM cats. 400x magnification. Scale bar = 50 μm.

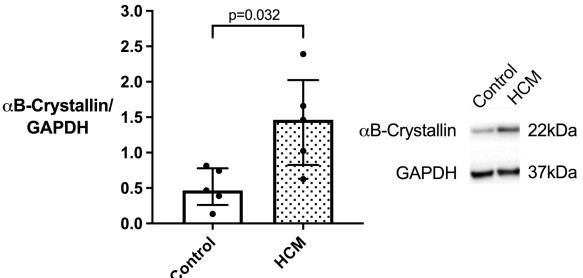

**Fig 9. An increase in expression of β-catenin in HCM cats was identified using immunoblotting.** Immunoblotting on LV extracts from control (white bar; n=5) and HCM (dotted bar; n=5) cats was carried out using a mouse monoclonal antibody against β-catenin. Bands at the expected molecular weight (92kDa) were detected using enhanced chemiluminescence. GAPDH was used as a loading control. The right shows a representative blot, while quantifications are shown on the left. A significantly greater quantity of β-catenin was identified in the HCM cats compared to controls. Data were analysed with Mann-Whitney test and graphed in bar chart format showing individual data points, median, 25th and 75th quartiles Refer to supplementary materials for full immunoblots.

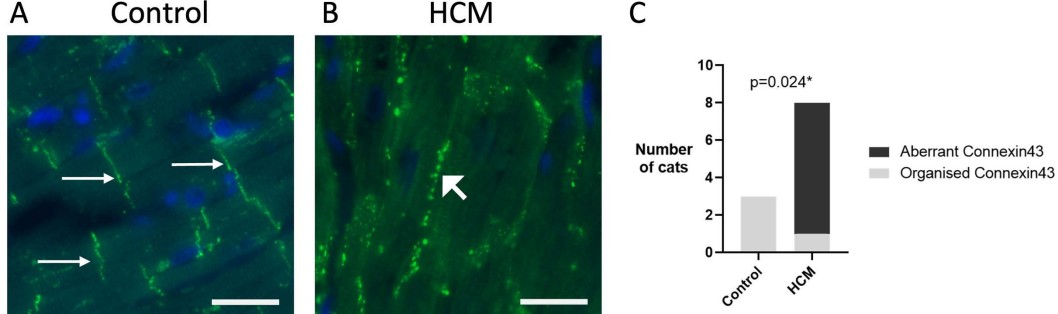

**Fig 10. Connexin-43 displays aberrant localisation in feline HCM.** A Connexin43 located at the intercalated discs (thin arrow) in the control cats. B Aberrantly localised connexin43 forming a line at the lateral side of the cardiomyocyte from a HCM heart (thick arrow) with reduced expression at the intercalated discs. C Number of cats in each group with or without aberrant localisation of connexin43 was shown in bar chart format. Disorganisation of connexin43 was exclusively found in feline HCM (p=0.024). Data was analysed with Fisher's exact test. 400x magnification. Connexin43 – green; Nuclei – blue; Scale bar = 25μm.

the ID. However, similar to our findings for β-catenin, marked abnormal localisation of connexin43 coincided with severely disrupted desmin alignment (Fig 11).

In summary, our results document disorganisation and aggregation of the IF desmin and its chaperone αB-crystallin in cardiomyocytes from HCM affected cats. Additionally, we have shown that desmin disorganisation at the intercalated disc is associated with altered cellular location of key junctional proteins which likely affects important cellular functions including cellular adhesion and electrical coupling.

## Discussion

This is the first study to identify aggregation of desmin and its chaperon protein αB-crystallin and the disruption to the ID proteins β-catenin, N-cadherin and connexin43 in cardiomyocytes from cats with HCM (Fig 12).

Our results significantly add to the current literature by confirming that IF disorganisation is a noteworthy finding in feline HCM with several important consequences including accumulation of toxic desmin/αB-crystallin aggregates which may suggest impaired PQC and abnormal localisation of junctional proteins required for correct functioning of the ID such as electrical coupling and intercellular linkage and communication.

The IF network together with its associated proteins, forms a dynamic framework that facilitates transmission of mechanochemical signals within cardiomyocytes as well as cross talk between organelles such as the sarcoplasmic reticulum and mitochondria [32,33,50,51]. Furthermore, the IF network extends to the costameres and ID to maintain cell adhesion, intercellular communication and electrical coupling [17,29,31–33]. Desmin is therefore vital for efficient functioning of crucial cellular processes and the disruption of the network seen in feline HCM will impair intra and intercellular performance.

Cytoplasmic aggregates are seen in desmin-related cardiomyopathies (DRM), caused by genetic mutations affecting desmin or its chaperone αB-crystallin [24,29,52]. These mutations result in formation of abnormally folded protein products and abnormal filament assemblies which are usually removed by the PQC, a process known as proteostasis [28,53–55]. Therefore, aggregate formation indicates overload of the PQC and failure to clear these terminally misfolded proteins [13,26,54]. Indeed, overload of the proteasome has been suggested as a more general mechanism to lead to a HCM phenotype, based on mouse models of sarcomeric and non sarcomeric HCM causing mutations [56,57]. It remains unclear whether it is the disruption of the IF network and its effect on interconnected proteins and organelles or the accumulation of toxic desmin aggregates per se, or a combination of both that is the primary mechanism driving cellular pathology [29,58,59]. However, desmin null mice or those with mutations either in desmin or αB-crystallin display IF disorganisation

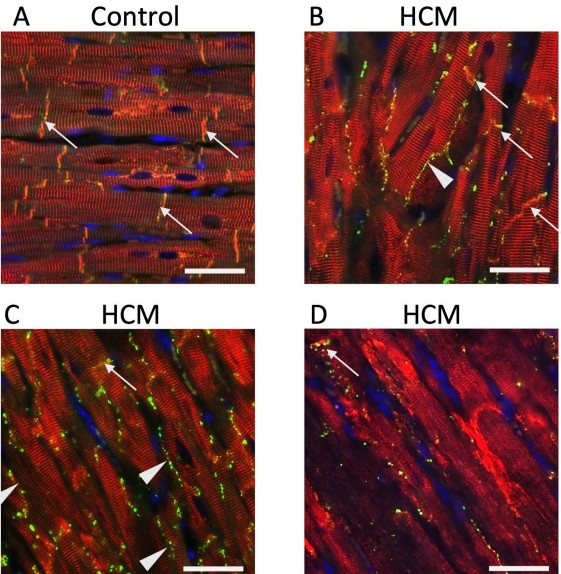

**Fig 11. Double immunostaining for desmin and connexin43 protein showed overlapping signal distribution at the intercalated discs in control cats but aberrant localisation in HCM cats.** A Connexin43 co-localised with desmin at the intercalated discs (arrows) in a control cat. B Disorganisation of some connexin43 to the lateral side of the cardiomyocytes (arrowheads) was identified while desmin was still co-expressed at the intercalated disc with connexin43 (arrows). C Desmin and connexin43 were less distinct at the intercalated discs (arrow) and most connexin43 localised to the lateral side of the cardiomyocytes (arrowheads). D In sections with severely disrupted alignment of desmin a similar abnormal localisation of connexin43 was observed. The arrow shows that connexin43 still co-localised with the desmin at that intercalated disc left. 400x magnification. Desmin – red; Connexin43 – green; Nuclei – blue; Scale bar = 25μm.

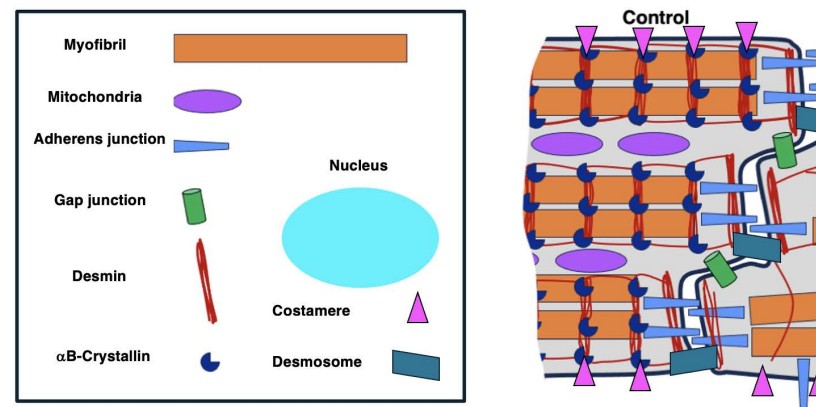

**Fig 12. Schematic illustration summarising the abnormal arrangement of desmin, αB-crystallin, and the junctional proteins in HCM affected cardiomyocytes.** In HCM affected hearts, desmin and αB-crystallin are lost from the Z-discs and intercalated discs. In some cardiomyocytes, aggregates containing desmin and αB-crystallin were observed, especially in the perinuclear area. While some cardiomyocytes retained the normal localisation of the junctional proteins β-catenin, N-cadherin, and connexin43, others showed disorganisation of these junctional proteins to the lateral side of the cardiomyocytes parallel to the orientation of the myofibrils. Note: The location of nucleus should be away from the intercalated disc and is drawn here only to indicate the perinuclear localization of the aggregates in the HCM cardiomyocytes.

and cardiac abnormalities prior to aggregate formation and present with pathology related to impaired function of cellular organelles including mitochondria [33,34,50,58,60–64]. Disruption of critical mitochondrial functions such as calcium handling, metabolism and programmed cell death, which are dependent upon an intact cytoskeletal network could provoke a myriad of pathological effects ultimately resulting in a cardiomyopathy phenotype [40,57,60,65–73].

Disruption of the IF network has been identified in human HCM patients with various patterns of expression noted in different cardiomyocytes from the same patient [13,14,74,75]. In cats with HCM, immunostaining of desmin was reported to be weaker than in controls at the Z discs and ID and granular staining within the cytoplasm of occasional cardiomyocytes which probably represents cytoplasmic accumulation of desmin was also described [30]. Additionally, accumulation of material, either described as electron dense rods extending from the Z-disc [2] or numerous thick clumps of Z-band material around the nucleus [76] has been reported in HCM cats although the composition of these materials was not confirmed. Our results demonstrate a similar heterogenous pattern of desmin staining as reported in cats and humans with HCM and expand on these earlier findings, by documenting extensive disruption of the desmin cytoskeletal network including at the Z discs and ID together with cytoplasmic accumulation of aggregates containing desmin and its chaperone αB-crystallin in cardiomyocytes from cats with HCM (Fig 2). Additionally, our cepstral analysis supports the disorganisation of the IF network in HCM cats identified by immunostaining which is important since a key aim of our study was to determine the impact of IF disorganisation on other key associated proteins (Fig 5). Furthermore, we show increased quantities of desmin and αB-crystallin in the HCM cats by western blot which may reflect overload of the PQC system and failure to remove the accumulated aggregates (Fig 3), however we did not perform transcriptional analysis on these samples to ascertain increased synthesis as an alternative explanation for increased protein levels. Furthermore, proteasome function tests were not evaluated in this study to directly quantify proteasome activity and protein quality control function. Interestingly, the double staining of desmin and αB-crystallin in left ventricular myocardium from HCM cats (Fig 4) suggests that the protein expression is dysregulated at a cell autonomous level. It is possible that the upregulation of αB-crystallin in the HCM hearts is a compensatory response to disruption of the desmin since by acting as a chaperone αB-crystallin facilitates removal of disrupted desmin by the PQC system [77]. Given the crucial roles of the cytoskeleton in normal mitochondrial function, the disruption of the IF network we identified is likely to compromise mitochondrial performance in HCM cats. Indeed, a transmission electron microscopy study revealed changes in mitochondrial localisation and impaired mitochondria function comprising reduced mitochondrial oxidative phosphorylation and increased reactive oxygen species release in cats with HCM [2]. Taken together these findings may suggest that disorganisation of the desmin network together with accumulation of toxic desmin rich aggregates potentially plays a significant role in the pathology of feline HCM [59].

Studies in human cardiomyopathy patients show that an additional consequence of IF disorganisation is abnormal localisation of important ID proteins including β-catenin, N-cadherin, desmoplakin and connexin43 which are involved in cell adhesion, mechanotransduction, cell-cell communication and electrical coupling [13–17,29,32,39,61,78–81].

We report for the first-time a complete absence and/or a diffuse pattern of expression of β-catenin and N-cadherin proteins at the ID and the displacement of β-catenin to the lateral side of cardiomyocytes in cats with HCM (Fig 6 and 8). The immunoblotting result suggests that the absence of β-catenin at the ID is not due to reduced protein expression. Furthermore, the results from our double immunostaining (Fig 7) suggest that aberrant β-catenin localisation appeared to occur following the loss of desmin at the intercalated discs which would be consistent with the premise that loss of IF organisation at the ID may potentially be responsible for disruption of key ID proteins.

We also report abnormal localisation of connexin43 in HCM cats with reduced expression at the ID and significant peripheral staining along the lateral sides of the cardiomyocyte (Fig 10). These findings are consistent with observations in human patients with HCM and restrictive cardiomyopathy [14,15,29]. A similar pattern of abnormal localisation has also recently been described in cats with HCM and restrictive cardiomyopathy [82]. Importantly our double immunostaining showed that the abnormal localization of connexin43 matched that of desmin suggesting a close association between the

two proteins as previously proposed based on immunoprecipitation assays in cardiomyocytes from human patients with chronic heart disease (Fig 11) [48]. However, without mechanistic experiments such as knockdown or rescue studies we cannot say for certain if the miss-location of these important junctional proteins is a consequence of desmin disruption. Interestingly, an early immunohistochemical study of connexin43 in Maine coon cats from a research colony with HCM due to the MYBPC3-A31P mutation revealed no changes in the localisation of connexin43 [83]. It is unclear whether this difference is associated with breed or the specific MYBPC3 mutation or related to the difference in degree of disease progression between the cats in our study and theirs.

Connexin43 is the main protein forming the gap junctions in ventricular cardiomyocytes which conduct ion currents between cardiomyocytes. Alterations in connexin43 distribution may therefore impair electrical conduction between adjacent cardiomyocytes. For instance, in two transgenic rabbit models of HCM, altered cellular connexin43 organization including lateralisation was associated with repolarisation abnormalities and increase arrhythmia propensity [84,85].

Sudden cardiac death (SCD) presumably due to a fatal arrhythmia is a common finding in cats with HCM and may in fact be the first manifestation of the disease [45]. Similarly, in the human population, HCM is considered the most common SCD-related cardiomyopathy and is one of the most common causes of SCD in young athletes [86]. Our finding of significantly altered distribution of connexin43 in cats with HCM may provide at least in part a molecular explanation for the high incidence of SCD in cats with this disease as has been suggested for human HCM [15].

The increased prevalence of male cats in the HCM group compared to control was not surprising as a male sex predisposition is well recognised in the feline population [7].

Our results identified abnormal aggregates comprising desmin and its chaperone αB-crystallin in cats with HCM. In addition, the abnormal localisation of key protein components of the ID in the HCM cats could impair the mechanical and electrical coupling in the heart (Fig 12). Together these effects may at least in part account for some of the clinicopathological findings characteristic of HCM including arrhythmogenesis and SCD. There are some limitations to our study. First, the immunohistochemistry and Western blots were performed on LV tissue taken at a single time point in the disease process for each cat and the variation in results between the HCM cats may represent different stages of disease progression or differences associated with specific causative mutations, epigenetic changes, or environmental influences between cats. Second, although the increased quantity of desmin and αB-crystallin in the HCM cats most likely results from reduced degradation due to impaired PQC, increased production cannot be ruled out especially in the absence of transcriptional data. Third, even though similar findings are reported in humans and other animal models, the number of cats included was small such that the immunostaining for proteins, such as N-cadherin and connexin43, might not be fully representative of HCM in this species. Fourth, although the age between the control cats and HCM cats was not significantly different, larger numbers of cats with a broad range of age should be used so as to critically appraise the possible impact of age on the function of PQC as aging has been associated with reduced capacity in PQC [87,88]. Additionally direct evaluation of proteasome function was not performed in this study. Fifth, post-translational modification of the proteins (desmin, αB-crystallin, β-catenin, N-cadherin, and connexin43), which would give a more in-depth understanding to how these proteins were regulated, was not investigated in this study. For example, in the case of desmin it was shown that phosphorylation by GSK3 is a key step for desmin aggregation and turnover [89,90]. Sixth, a key protein component of the desmosome, desmoplakin, was not evaluated due to our inability to find an antibody that recognised the feline protein. Unfortunately, this is a common problem when working with feline tissue and precluded the assessment of all the main proteins at the ID. Seventh, it is important to emphasise that the cellular changes described here are not unique to HCM but can occur in other forms of cardiac disease [17,19,20].

## Supporting information

**S1 Table. Clinical data, echocardiographic parameters, and histopathology diagnosis.**
(DOCX)

**S2 Table. Clinical presentation and echocardiography summary.**
(DOCX)

**S3 Table. Information on echocardiographic views and procedures.**
(DOCX)

## Author contributions

**Conceptualization:** Wan-Ching Cheng, Virginia Luis Fuentes, Elisabeth Ehler, David J Connolly.

**Data curation:** Wan-Ching Cheng, Lois Wilkie, Melanie Dobromylsky, Mark R Holt, Elisabeth Ehler.

**Formal analysis:** Wan-Ching Cheng, Charlotte Lawson, Mark R Holt, Elisabeth Ehler.

**Funding acquisition:** Wan-Ching Cheng, David J Connolly.

**Investigation:** Wan-Ching Cheng, Charlotte Lawson, Lois Wilkie, Melanie Dobromylsky, Mark R Holt, Elisabeth Ehler, David J Connolly.

**Methodology:** Wan-Ching Cheng, Charlotte Lawson, Lois Wilkie, Melanie Dobromylsky, Elisabeth Ehler, David J Connolly.

**Project administration:** David J Connolly.

**Resources:** Melanie Dobromylsky, Elisabeth Ehler, David J Connolly.

**Supervision:** Charlotte Lawson, Virginia Luis Fuentes, Elisabeth Ehler, David J Connolly.

**Writing – original draft:** Wan-Ching Cheng.

**Writing – review & editing:** Virginia Luis Fuentes, Mark R Holt, Elisabeth Ehler, David J Connolly.

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
