## [Decision Letter · Decision Letter 0]

PONE-D-25-16588Desmin disorganisation: a key feature in feline hypertrophic cardiomyopathyPLOS ONE

Dear Dr. Connolly,

Thank you for submitting your manuscript to PLOS ONE. After careful consideration, we feel that it has merit but does not fully meet PLOS ONE’s publication criteria as it currently stands. Therefore, we invite you to submit a revised version of the manuscript that addresses the points raised during the review process.

We look forward to receiving your revised manuscript.

Kind regards,

Andreas Brodehl, Ph.D.

Academic Editor

PLOS ONE

Journal Requirements:

3. To comply with PLOS ONE submissions requirements, in your Methods section, please provide additional information regarding the experiments involving animals and ensure you have included details on (1) methods of sacrifice, (2) methods of anesthesia and/or analgesia, and (3) efforts to alleviate suffering.

Reviewers' comments:

Reviewer's Responses to Questions

**Comments to the Author**

1. Is the manuscript technically sound, and do the data support the conclusions?

Reviewer #1: Yes

Reviewer #2: Yes

Reviewer #3: Yes

Reviewer #4: Partly

Reviewer #5: Yes

Reviewer #6: Yes

2. Has the statistical analysis been performed appropriately and rigorously? 

Reviewer #1: Yes

Reviewer #2: Yes

Reviewer #3: N/A

Reviewer #4: Yes

Reviewer #5: Yes

Reviewer #6: I Don't Know

3. Have the authors made all data underlying the findings in their manuscript fully available?

Reviewer #1: No

Reviewer #2: Yes

Reviewer #3: Yes

Reviewer #4: No

Reviewer #5: Yes

Reviewer #6: Yes

4. Is the manuscript presented in an intelligible fashion and written in standard English?

Reviewer #1: Yes

Reviewer #2: Yes

Reviewer #3: Yes

Reviewer #4: Yes

Reviewer #5: Yes

Reviewer #6: Yes

5. Review Comments to the Author

Reviewer #1: I have reviewed a manuscript by Wan-Ching Cheng et al. investigating changes of desmin and associated proteins in cats with hypertrophic cardiomyopathy.

This is a well written manuscript with a carefully conducted study, a good number of animal samples were used. However, the investigations are somewhat descriptive. I cannot see an immediate benefit to veterinary practice.

A few points should be addressed before publication:

1.) How specific are the changes for Hypertrophic Cardiomyopathy in cats. Could they be a hallmark for any type of heart failure? What is known in the literature?

2.) Introduction goes straight into molecular changes. What are symptoms of HCM (in humans and cats) and how is HCM diagnosed in cats? What is cardiac T-fast? Please explain.

3.) The references to Protein Quality Control are speculative and should be toned down throughout the manuscript in the absence of data on activity of the Ubiquitin Proteasomal System and Autophagy

4.) I assume genetic testing is not done routinely in cats. Could the authors give an overview which HCM disease genes/variants are common in cats of various breeds? Would it make sense to identify pure breeds (likely to be inbred) and outbred cats int table S1/S2?

5.) Figures 1 and 13 should also feature desmosomes

6.) Check immunofluorescence figures for sufficient resolution (e.g. Figs 2 and 5 look blurry)

7.) Figure 3 (Western blots) why are only few samples used here, but much higher n numbers in Figure 10? Show full blots with all samples in Suppl. Material. for all western blots.

Figure 3B swap alpha-beta crystalline (to go on top) with Gapdh

8.) Figure 6: label y-axis, show individual data points per animal?

9.) Figure 12: label what is shown in green and red

Reviewer #2: The manuscript titled "Desmin disorganisation: a key feature in feline hypertrophic cardiomyopathy" presents compelling evidence that intermediate filament (IF) disruption, particularly the aggregation of desmin and αB-crystallin, constitutes a significant pathological feature of feline hypertrophic cardiomyopathy (HCM). The authors provide robust histological and biochemical data, supported by cepstrogram analysis and immunostaining, which align the features of feline HCM with established human pathomechanisms. The paper is generally well-structured, and the findings possess considerable translational relevance.

However, several critical comparisons (e.g., desmin, αB-crystallin, connexin43) are based on limited and inconsistent sample sizes across groups. The authors should provide a rationale for the allocation of samples across assays and to clarify the measures taken to ensure statistical robustness.

In addition to cepstrogram analysis and qualitative immunostaining, have the authors quantified desmin disorganization (e.g., through fluorescence intensity or spatial metrics)? If such quantification has been conducted, it should be included in the manuscript; if not, the authors should consider whether such quantification would enhance the interpretation of their data.

The Western blot results indicate an accumulation of desmin and αB-crystallin. Did the authors assess ubiquitin tagging or proteasome function to substantiate claims of protein quality control dysfunction? It would be beneficial to evaluate proteasome markers.

The discussion addresses potential mitochondrial dysfunction related to IF disruption. Have the authors considered employing transmission electron microscopy (TEM) or immunostaining for mitochondrial proteins such as VDAC or TOM20 in these samples?

Were any of the HCM-affected cats genotyped (e.g., for the MYBPC3-A31P mutation in Maine Coons)? Genetic stratification could elucidate the inter-individual variability observed in desmin pathology and connexin43 dislocation.

While the abnormal localization of connexin43 suggests electrical uncoupling, the authors should consider incorporating electrophysiological assessments (e.g., ECG or optical mapping) to establish a connection between these findings and arrhythmic risk. The study would significantly benefit from functional readouts, such as calcium handling, contractility, or conduction velocity, particularly to support the mechanistic link to sudden cardiac death.

Reviewer #3: Critical Evaluation of ‘Desmin disorganization: a key feature in feline hypertrophic cardiomyopathy’

The article presents an analysis of cytoskeletal disruption, mostly desmin disorganization and its impact of chaperone and structural proteins in cardiomyocytes in feline hypertrophic cardiomyopathy tissues.

Although, the co-localization of desmin and its chaperone (αB-crystallin) and ID proteins such as β-catenin and connexin43 were discussed and studied, it remains unclear whether desmins disruption drive the misslocation of the mentioned proteins. Mechanistic experiments such as knockdown or rescue studies would have reinforce the connection between desmin, αB-crystallin, β-catenin and connexin43. Furthermore, due to the results of western blot experiments, the quantitative level of the proteins were discussed without having transcriptional data to rule out increased synthesis. Next, the sex bias of investigated cats were mentioned (p=0.07) but without further discussion. A genotype analysis was not performed for a further study of genetic factors of HCM.

Despite these limitations, the article contribute a feline perspective on HCM research.

Reviewer #4: In the manuscript the authors describe that in feline HCM- similiar to human HCM- aggregates of Desmin an alphaB-crystalline may occur . In addition they also observed a disorganization of components of adherens and gap junctions, which suggest an impaiered cell-cell communication. Since there is not much known about it in cats, this is an important study , though there are some majors concerns, listed below.

I appreciated that in the discussion sections the limitations of this study are described. I But I also would have preferred some information on the cats specifities, in case of anomalous findings. I had problems to access Fig. 13 Only a fast glimpse was possible and I did not find any full blot images in the supplements as was announced.

Major concerns:

1) Line 51 ff. What is meant by “comparable“ the statement a completely refer to human CM. What about feline CM. Is there know anything know about IF dysfunctuntion/disorganization in cats Please distinguish and describe clearly the situation in cats vs human.

2) Fig.1 it would be grat, if you could also mark costameres, desmosomes and th and gap/ adherens junctions in the summary. Not all readers are familiar with this. In general give at much information as possible in the figures as they should be self-explanatory.

3) Line 86ff. This paragraph is not very clear. It sound as if ECGs and cardiac TFASts have been performed with dead cats. Cardiac T-fast to my knowledge is an expression for tachycardia and cardiac TFAST an ultra sonic examination method, which should be explained shortly (e.g.alternating left and right lateral recumbency / sternal recumbency, image optimization)

4) Line 99ff why ony some cats were selected exclusively for western blotting and others only for immunofisochemistry, why not all for both? Do controls and hcm slected tissue match in age, gender.

5) Lines211ff does it make sense to calculate a median age, since the range is very large. Wouldn’t it be better to select 2 different age groups e.g ≤7 and >7?

6) Isn‘t it possible to combine some figures into one e.g. Fig 3 and 4 For an easier understanding please mark the axis of fig 6 e.g. with degree of disorder or something similar

7) Line 275 f which oft he cats investigated had a severe disruption of desmin architecture: are thes cats older? Are male or female? Specific race, differences in HCM progression etc? should be discussed

8) Legend to Fig 9 concerningthe 3 cats where no lateralization was observed see comment 7

9) Line 424 which post-translational modifications you are thinking of ( oxidative modifications, phosphprylation)

10) Which consequences do the results described here could have for treatment/management of feline HCM what are the furure aspects?

Minor comments

1) A few typing mistakes: Line 76: … ß-catenin, and intracellular…; line 104: desribed; line 244: ,respectively

2) Supplement table 2: in the legend FS is defined, which is not found in the table only FN or F

3) Line 74 please define freewallk ( part of ventricle wall not connected to septum or apex?)

4) Shouldn’t

5) Line 288: …. Expression of ß-catenine observed by ?immunostaining.

6) Line 179 figure heading shouldn’t cepstrogram analysis be replaced by cepstral analysis since cepstrogram ist he result of cepstarl analysis?

7) Line 292 heading fig.10 : expression cannot be increased by immuno blotting

Reviewer #5: Hypertrophic cardiomyoathy (HCM) in human and rodents is characterized by an increase in wall thickness and myocyte disarray. HCM is often accompanied by alterations in the intracellular organisation of the intermediate filament system formed by desmin. The authors demonstrate that similar alterations of desmin organisation can also be observed in feline HCM cardiomyocytes. Desmin disorganisation can result from a number of different causes: genetic alterationsm of desmin itself or of desmosome components or an inadequate protein quality control system.

The authors demonstrate that feline hearts affected by HCM show phenotypic changes similar to reported alteration in human or rodent HCM hearts. They identify a disorganised distribution of desmin and the formation of large intra-myocyte aggregates composed of desmin and its chaperone αB-crystalline. Since Western blots indicate an intracellular increase in desmin, it is suggested that the protein quality control system is overloaded.

The disorganisation of desmin is probably causing a disorganization of junctional complexes leading to aberrant localisation of a number of proteins normally residing Z-discs and intercalated discs, such connexin 43, N-cadherin and β-catenin localizing at the lateral sides of HCM cardiomyocytes.

The data are well documented and clearly presented and fully merit acceptance for publication.

Minor points:

In the available version were no Figure legends.

Therefore it was not possible to understand Figure 6, since no information about the meaning of the y-axis was given. In particular, the aim of the procedure used (Cepstrogram) appeared unclear. Was it a particular structure (desmin disorganization and aggregates) of the images to be analysed?

There are some typos.

Reviewer #6: Based in published data that cytoplasmic aggregates of the intermediate filament desmin and its chaperone heat shock protein αB-crystallin together with disorganisation of associated junctional proteins including β-catenin, N-cadherin and connexin43 are identified in cardiomyocytes from human HCM patients and mouse models in the present study Wan-Ching Cheng and et al. sought to identify and further characterise the localisation of these proteins in a cohort of cats with HCM. They used a pool of 12 control and 23 HCM feline cardiac samples for their analysis.

By immunostaining they showed similar aggregate formation and/or mislocalization of these proteins in feline HCM cardiomyocytes and by western blot increased expression of desmin, αB-crystallin and N-cadherin. Their data are interesting and are suggestive of common subcellular responses of cardiomyocytes to HCM in mammals.

I have some comments for the presented data:

• First, I think the group of the HCM animals lacks information. As indicated for the control samples (used in another publication (doi.org/10.3390/ani13132112), representative histopathological features of HCM feline hearts (HE, Masson's Trichrome…) should be presented. It will be informative to have the same area by histological staining next to the immunofluorescence stained samples.

• Is genetic analysis for HCM samples available or already published? (doi.org/10.1038/s41598-025-87852-5). It will be nice to be included.

• Figure 2 (legend) says: “Confocal micrographs of feline hearts” As indicated in methods “Image analysis. Fluorescent labelling of the proteins was detected using a Leica DMRA2 upright microscope. Is not confocal microscope and I think for this type of analysis confocal microscope analysis is needed as photos are blurry! For the reagent control sample a control of HCM sample should be also included.

• Some figures I think could be combined as for example figure 1 and 13, figure 2 and 5, figure 3 and 4, figures 11 and 12

• As the authors mention in the limitation section, post-translational modifications (PMT) of the proteins analysed would give a more in depth understanding to how these proteins are regulated. Regarding desmin, cleavage of desmin could be a PMT linked to aggregate formation and or mislocalization (DOI: 10.1002/humu.20941; doi:10.1093/cvr/cvu003; doi: 10.1096/fj.07-088724). Over exposure of desmin western blot (figure 3) could be informative.

• As a lot of the HCM animals had Hind limb paresis it will be nice to include a desmin immunostaining of skeletal muscle tissue section.

6. PLOS authors have the option to publish the peer review history of their article (what does this mean? ). If published, this will include your full peer review and any attached files.

**Do you want your identity to be public for this peer review?** For information about this choice, including consent withdrawal, please see our Privacy Policy .

Reviewer #1: No

Reviewer #2: **Yes: ** Nazha Hamdani

Reviewer #3: No

Reviewer #4: No

Reviewer #5: No

Reviewer #6: No

---

## [Author Response · Author response to Decision Letter 1]

18 Jun 2025

Dear Reviewers we would like to thank you for your considered and helpful comments regarding this manuscript. We hope we have been able to answers all your queries in a satisfactory way and we appreciate that the manuscript is much improved following your comments.

Reviewer #1:

I have reviewed a manuscript by Wan-Ching Cheng et al. investigating changes of desmin and associated proteins in cats with hypertrophic cardiomyopathy.

This is a well written manuscript with a carefully conducted study, a good number of animal samples were used. However, the investigations are somewhat descriptive. I cannot see an immediate benefit to veterinary practice. We thank the reviewer for taking the time to assess our manuscript. We agree that this work is unlikely to result in an immediate benefit to veterinary practice. However, exploring fundamental cellular alterations in cardiomyocytes from HCM affected cats is an important step to help understand potential pathological principles. For instance, desmin disruption may lead to mal location of connexin43 resulting in malignant arrhythmia and sudden cardiac death seen in cats with HCM. Given that PLOS ONE is not an exclusively clinically based Journal we felt this manuscript aligned well with the broader themes of the journal.

A few points should be addressed before publication:

1.) How specific are the changes for Hypertrophic Cardiomyopathy in cats. Could they be a hallmark for any type of heart failure? What is known in the literature? Thank you for this comment, we agree the changes we describe are not unique to HCM in humans and as we pointed out in the introduction (Lines 62-64) have been identified in other cardiac diseases including dilated cardiomyopathy, mitral valve disease and as a result of desmin mutations. That said, the overwhelming cause of feline heart disease is HCM. We have added a sentence to the limitations section to emphasise this point. LINES:486-488 untracked manuscript file “Seventh, it is important to emphasise that the cellular changes described here are not unique to HCM but can occur in other forms of cardiac disease (17,19, 20).”

2.) Introduction goes straight into molecular changes. What are symptoms of HCM (in humans and cats) and how is HCM diagnosed in cats? What is cardiac T-fast? Please explain. In response to you comment we have added a brief paragraph outlining the prevalence and clinical signs in feline HCM as well as diagnostic methods. LINES: 44-58 untracked manuscript file “Hypertrophic cardiomyopathy (HCM) is a serious disease in humans and cats and exhibits considerable similarities at the molecular, cellular and whole organ levels [1-5]. It is characterized by a hypertrophied and non-dilated left ventricle in the absence of abnormal loading conditions capable of producing left ventricular hypertrophy (Luis Fuentes et al 2020). HCM is the most common heart disease in cats affecting up to 15% of the general feline population (Payne JR et al 2015). Echocardiography is the clinical gold standard for HCM diagnosis in cats, and the most used imaging modality in humans (Luis Fuentes et al 2020, Elliott P et al 2014). Clinical signs in cats include respiratory distress due to congestive heart failure, pelvic limb ischaemic paralysis/paresis due to aortic thromboembolism or sudden arrhythmogenic death. Prognosis for cats with HCM is very variable, with median survival times ranging from 596 to 1276 days (Novo Matos J 2023). The role of genetic testing is very limited in cats as to date only two disease associated variants in MYBPC3 have been identified, one in the Maine coon and one in the Ragdoll breed; additionally, a potentially pathogenic variant was found in MYH7 in a domestic shorth air (Raffle J et al 2025).”

We have used the abbreviation T-fast but based on a comment from another reviewer the correct term TFAST should be used. TFAST is a rapid thoracic ultrasound examination used in an emergency situation to evaluate cardiac, pulmonary, mediastinal and pleural structures. This will enable the clinician to identify the presence of obvious cardiac abnormalities such as HCM and atrial thrombi as well as abnormal fluid accumulation such as pleural effusion or pulmonary oedema indicative of congestive heart failure. We have updated our abbreviation throughout and explained the technique in the M+M section. LINES: 107-110 untracked manuscript file “Complete echocardiographic examination or Cardiac TFAST (a rapid thoracic ultrasound examination used in an emergency situation with the animal generally in sternal recumbency optimising the image to evaluate cardiac, pulmonary, mediastinal and pleural structures) was performed in 11/12 control group cats.”

3.) The references to Protein Quality Control are speculative and should be toned down throughout the manuscript in the absence of data on activity of the Ubiquitin Proteasomal System and Autophagy. Thank you for this comment, we agree and have made suitable changes throughout the manuscript.

4.) I assume genetic testing is not done routinely in cats. Could the authors give an overview which HCM disease genes/variants are common in cats of various breeds? You are correct very limited genetic testing is performed in cats as a result of the sparsity of information about disease causing variants. To date two disease associated variants in MYBPC3 have been identified, one in the Maine coon and one in the Ragdoll breed. doi: 10.1016/j.ygeno.2007.04.007, doi: 10.1093/hmg/ddi386. Additionally, a potentially pathogenic variant was found in MYH7 in a domestic shorthair. We have added a sentence to the introduction to this effect LINES: 53-58 untracked manuscript file “The role of genetic testing is very limited in cats as to date only two disease associated variants in MYBPC3 have been identified, one in the Maine coon and one in the Ragdoll breed; additionally, a potentially pathogenic variant was found in MYH7 in a domestic shorth air (Raffle J et al 2025).”

Would it make sense to identify pure breeds (likely to be inbred) and outbred cats int table S1/S2? In table S1 we identify which cats are outbred and which are purebred. All cats designated DHS (domestic shorthair) are outbred, we have added a sentence to the table legend to clarify this.

5.) Figures 1 and 13 should also feature desmosomes. Thank you for this comment, the figures have been updated in line with your request.

6.) Check immunofluorescence figures for sufficient resolution (e.g. Figs 2 and 5 look blurry) Thank you for the comment - We agree that on the submitted build PDF the figures do look blurry as shown by the indistinct labelling. This blurriness is not present on our actual figures and therefore likely is an issue with the PDF building programme – I will defer to the Editor.

7.) Figure 3 (Western blots) why are only few samples used here, but much higher n numbers in Figure 10? We thank the reviewer for this helpful comment. The difference in sample numbers reflects the sequential development of the study. Desmin was initially analysed in 5 HCM and 5 control cats, which was the pre-defined scope of the original investigation focused on intermediate filament disruption. Upon observing striking findings in desmin expression, we decided to further examine its chaperone protein αB-crystallin in the same group of animals to assess the relationship between their expression levels.

Later, we sought to explore junctional proteins and analysed β-catenin using sample resources from a parallel study on myocardial fibrosis (reference 37). This set included five additional cats per group, resulting in a total of 10 HCM and 10 control cats, and enabled an expanded analysis of β-catenin expression.

We have clarified these experimental phases and sample numbers in the revised Materials and Methods section LINES: “Five control and 5 HCM cats were used for immunoblotting of desmin and αB-crystallin; and 10 control and 10 HCM cats were used for junctional protein β-catenin immunoblotting. αB-crystallin was analysed in the same set of cats as used for desmin to assess paired expression. β-catenin was analysed using membrane resources from a parallel study on myocardial fibrosis (reference 37), which included five additional cats in each group.”

Show full blots with all samples in Suppl. Material. for all western blots. Our apologies if these full blots were not available to you – again they were uploaded when we build the PDF – I will re-upload and defer to the Editor if they do not appear again

Figure 3B swap alpha-beta crystalline (to go on top) with Gapdh. Thank you for your helpful suggestion. We have updated Figure 3B to reflect the correct arrangement, with αB-crystallin now shown on top and GAPDH (loading control) below, as per standard western blot presentation format.

8.) Figure 6: label y-axis, show individual data points per animal? We have amended the figure as requested.

9.) Figure 12: label what is shown in green and red Thank you for the comment in our legends we clarify which protein is labeled red and which green. For figure 8: Desmin – red; β-catenin – green; Nuclei – blue. For figure 12: Desmin – red; Connexin43 – green; Nuclei – blue

Reviewer #2:

The manuscript titled "Desmin disorganisation: a key feature in feline hypertrophic cardiomyopathy" presents compelling evidence that intermediate filament (IF) disruption, particularly the aggregation of desmin and αB-crystallin, constitutes a significant pathological feature of feline hypertrophic cardiomyopathy (HCM). The authors provide robust histological and biochemical data, supported by cepstrogram analysis and immunostaining, which align the features of feline HCM with established human pathomechanisms. The paper is generally well-structured, and the findings possess considerable translational relevance. We thank the reviewer for their supportive words

However, several critical comparisons (e.g., desmin, αB-crystallin, connexin43) are based on limited and inconsistent sample sizes across groups. The authors should provide a rationale for the allocation of samples across assays and to clarify the measures taken to ensure statistical robustness. We thank the reviewer for this valuable comment. The discrepancies in sample numbers across assays were due to a combination of practical factors inherent to working with clinical samples rather than standardized laboratory models. Our initial experimental design targeted 5 HCM and 5 control cats for each analysis. However, due to tissue limitations (e.g., exhausted paraffin blocks), loss of slides during shipping, occasional failed staining, and limited availability of some antibody reagents, we supplemented with additional cases when possible.

Furthermore, control hearts are inherently more difficult to obtain compared to clinical HCM cases, especially when histologically confirmed as normal. This contributed to some differences in group sizes.

We acknowledge that our initial table included incorrect n values (Table 1), which have now been corrected. Despite these variations, all statistical comparisons (when performed) were performed using appropriate non-parametric tests suited for small and unbalanced groups based on discussions with our College statistician.

In addition to cepstrogram analysis and qualitative immunostaining, have the authors quantified desmin disorganization (e.g., through fluorescence intensity or spatial metrics)? If such quantification has been conducted, it should be included in the manuscript; if not, the authors should consider whether such quantification would enhance the interpretation of their data. Thank you for this insightful comment. Our cepstral analysis was derived as a way of determining the desmin organization as this is key to one of the central themes of the paper that desmin disorganisation impacts other important adhesion proteins etc. We did not quantify the degree of desmin disorganisation, but this is something that could be considered in future studies.

The Western blot results indicate an accumulation of desmin and αB-crystallin. Did the authors assess ubiquitin tagging or proteasome function to substantiate claims of protein quality control dysfunction? It would be beneficial to evaluate proteasome markers. Thank you for this comment we agree that this would have been very interesting to perform but was beyond the scope of the PhD which forms the basis of this work. Additionally in line with a similar comment from Reviewer 1 we have altered the discussion and added to the limitations section to reflect that this study has not confirmed that dysfunction of protein quality control was the cause of the changes to desmin quantification we identified and other possible explanations such as increased synthesis was not ruled out in this study.

The discussion addresses potential mitochondrial dysfunction related to IF disruption. Have the authors considered employing transmission electron microscopy (TEM) or immunostaining for mitochondrial proteins such as VDAC or TOM20 in these samples? We did not perform this analysis but again we thank the reviewer for these excellent suggestions which hopefully will form the basis of future work in this exciting area.

Were any of the HCM-affected cats genotyped (e.g., for the MYBPC3-A31P mutation in Maine Coons)? Genetic stratification could elucidate the inter-individual variability observed in desmin pathology and connexin43 dislocation. Genotyping was not performed in any of the cats including the two Maine Coons – please refer to our response to Reviewer 1 regarding the status of genotyping in diagnosis of feline HCM. Specifically regarding the MYBPC3-A31P mutation - the picture even within the Maine coon breed is mixed with both genotype +ve/phenotype -ve and genotype -ve/phenotype +ve individuals (J Vet Cardio. 2010 Dec;12(3):155-61. doi: 10.1016/j.jvc.2010.06.004)

While the abnormal localization of connexin43 suggests electrical uncoupling, the authors should consider incorporating electrophysiological assessments (e.g., ECG or optical mapping) to establish a connection between these findings and arrhythmic risk. The study would significantly benefit from functional readouts, such as calcium handling, contractility, or conduction velocity, particularly to support the mechanistic link to sudden cardiac death. Again, we thank the reviewer for these excellent suggestions. Unfortunately, ECG data was not available for most of the HCM cats as they invariably presented to our out of hours emergency service. However, none experienced sudden cardiac death and rather were euthanised for the reasons outlined in Table S2. Arrhythmias were detected by auscultation in 6/23 cats but these were intermittent and not considered haemodynamically significant. ECGs were performed in two cats that were subsequently transferred to the Cardiology service (HCM 11 and HCM 22) in both cats, intermittent uniform ventricular premature complexes were identified. Regarding optical mapping this is not a technique we have available in our institution and given the cardiac tissue used in this study came from clinical cases (not laboratory animals) the logistics associated with collaborating with another institution to perform OM would be hugely challenging.

Reviewer #3:

Critical Evaluation of ‘Desmin disorganization: a key feature in feline hypertrophic cardiomyopathy’

The article presents an analysis of cytoskeletal disruption, mostly desmin disorganization and its impact of chaperone and structural proteins in cardiomyocytes in feline hypertrophic cardiomyopathy tissues.

Although, the co-localization of desmin and its chaperone (αB-crystallin) and ID proteins such as β-catenin and connexin43 were discussed and studied, it remains unclear whether desmins disruption drive the misslocation of the mentioned proteins. Mechanistic experiments such as knockdown or rescue studies would have reinforce the connection between desmin, αB-crystallin, β-catenin and connexin43. We thank the reviewer for their input into our manuscript. We agree with the Reviewer and accept that this is primarily a descriptive study that will hopefully lead to further investigation. The further studies you suggest were beyond the scope of this initial study however, we have a

---

## [Editor Report · Decision Letter 1]

Desmin disorganisation: a key feature in feline hypertrophic cardiomyopathy

PONE-D-25-16588R1

Dear Dr. Connolly,

We’re pleased to inform you that your manuscript has been judged scientifically suitable for publication and will be formally accepted for publication once it meets all outstanding technical requirements.

Kind regards,

Andreas Brodehl, Ph.D.

Academic Editor

PLOS ONE
---

## [Editor Report · Acceptance letter]

PONE-D-25-16588R1

PLOS ONE

Dear Dr. Connolly,

I'm pleased to inform you that your manuscript has been deemed suitable for publication in PLOS ONE. Congratulations! Your manuscript is now being handed over to our production team.

Kind regards,

on behalf of

Dr. Andreas Brodehl

Academic Editor

PLOS ONE